# Comprehensive approach to HIV/AIDS testing and linkage to treatment among men who have sex with men in Curitiba, Brazil

Marly Marques da Cruz[1], Vanda Lúcia Cota[1]*, Nena Lentini[2], Trista Bingham[3], Gregory Parent[3], Solange Kanso[1], Liza Regina Bueno Rosso[4], Bernardo Almeida[5], Raquel Maria Cardoso Torres[1], Cristiane Yumi Nakamura[4], Ana Carolina Faria e Silva Santelli[6]

1 Sergio Arouca National School of Public Health, Oswaldo Cruz Foundation, Rio de Janeiro, Brazil,
2 Revolving Fund for Vaccine Procurement, Pan American Health Organization, Washington, DC, United States of America, 3 Division of Global HIV and TB (DGHT), Centers for Disease Control and Prevention (CDC), Atlanta, Georgia, United States of America, 4 Municipal Health Secretariat of Curitiba, Curitiba, Brazil, 5 Department of Hospital Epidemiology, Federal University of Paraná, Curitiba, Brazil, 6 Division of Global HIV and TB (DGHT), Centers for Disease Control and Prevention (CDC), Country Office in Brasília, Brasília, Brazil

☯ These authors contributed equally to this work.
* vanda.cota@ensp.fiocruz.br

## Abstract

### Introduction

The Curitiba (Brazil)-based Project, *A Hora é Agora* (AHA), evaluated a comprehensive HIV control strategy among men who have sex with men (MSM) aimed at expanding access to HIV rapid testing and linking HIV-positive MSM to health services and treatment. AHA's approach included rapid HIV Testing Services (HTC) in one mobile testing unit (MTU); a local, gay-led, non-governmental organization (NGO); an existing government-run health facility (COA); and Internet-based HIV self-testing. The objectives of the paper were to compare a) number of MSM tested in each strategy, its positivity and linkage; b) social, demographic and behavioral characteristics of MSM accessing the different HTC and linkage services; and c) the costs of the individual strategies to diagnose and link MSM to services.

### Methods

We used data for 2,681 MSM tested at COA, MTU and NGO from March 2015 to March 2017. This is a cross sectional comparison of the demographics and behavioral factors (age group, race/ethnicity, education, sexually transmitted diseases, knowledge of AHA services and previous HIV test). Absolute frequencies, percentage distributions and confidence intervals for the percentages were used, as well as unilateral statistical tests.

### Results and discussion

AHA performed 2,681 HIV tests among MSM across three in-person strategies: MTU, NGO, and COA; and distributed 4,752 HIV oral fluid tests through the self-testing platform. MTU, NGO and COA reported 365 (13.6%) HIV positive diagnoses among MSM, including

**Data Availability Statement:** All relevant data are within the manuscript.

**Funding:** This publication was supported by the Cooperative Agreement Number NU2G GH001152, funded by the United States President's Emergency Plan for AIDS Relief (PEPFAR) through the Centers for Disease Control and Prevention (CDC). Its contents are solely the responsibility of the authors and do not necessarily represent the official views of the CDC or the Department of Health and Human Services (HHS).

**Competing interests:** All authors have declared that no competing interests exist.

28 users with previous HIV diagnosis or on antiretroviral treatment for HIV. Of these, 89% of MSM were eligible for linkage-to-care services. Linkage support was accepted by 86% of positive MSM, of which 66.7% were linked to services in less than 90 days. The MTU resulted in the lowest cost per MSM tested ($137 per test), followed by self-testing ($247).

## Conclusions

AHA offered MSM access to HTC through innovative strategies operating in alternative sites and schedules. It presented the Curitiba HIV/AIDS community the opportunity to monitor HIV-positive MSM from diagnosis to treatment uptake. Self-testing emerged as a feasible strategy to increase MSM access to HIV-testing through virtual tools and anonymous test kit delivery and pick-up. Cost per test findings in both the MTU and self-testing support expansion to other regions with similar epidemiological contexts.

## Introduction

Although the number of new HIV infections is declining globally, the number of people living with HIV (PLHIV) is still high. Over 37 million people are estimated to be living with HIV around the globe, while 1.7 million new infections and 770,000 AIDS-related deaths were reported in 2018 [1]. Continued progress in treatment has resulted in decreased AIDS-related mortality and increased life expectancy among PLHIV. However, late HIV diagnosis remains as a key challenge to epidemic control. In 2017, an estimated 16% of PLHIV were unaware of their serological status [2, 3]. Thus, periodic HIV testing services (HTC) and early linkage of positive cases to the health system [4] are critical prevention and treatment strategies to strengthen the global epidemic response.

Brazil has become a model in HIV control thanks to adoption of progressive policies and continued investments in prevention and treatment over the past 35 years. However, the country's ability to control the epidemic among men who have sex with men (MSM) remains a challenge given an increasing trend in HIV diagnoses among males ages 15 to 24 years [5, 6]. AIDS detection rates in Brazil in 2019 was 17.8 cases per 100,000 inhabitants and it varies widely among state capitals. Among men, the AIDS detection rate was 25.2 cases, and an increase was observed in the groups mainly between 15 and 19 years and 20 to 24 years, which were, respectively, 64.9% and 74.8% [7].

Studies conducted in the past 20 years indicate that MSM report higher HIV detection rates when compared to the general population [8]. Thus, MSM became the key population for HIV control by the Government of Brazil (GOB), which recognized that HIV-infected MSM who know their status are able to initiate treatment early to reduce forward transmission [9–11].

Curitiba is the capital of the southern Brazilian state of Paraná. With an estimated population of 1.9 million (approximately 500,000 between 18 to 29 years, 79% White), Curitiba ranks 5th in terms of *per capita* income among Brazilian municipalities and reports average schooling completion of 12 years [12]. The local basic health care network includes 113 facilities.

The AIDS detection rate in Curitiba was 13.8 cases per 100,000 inhabitants and the sex ratio was 4.0 (M:W) 40 men for every 10 women. Curitiba has observed increased detection of HIV among men ages 20 to 29 years and a higher proportion of all HIV cases attributed to male-to-male sexual transmission [8, 13]. In 2014, with the goal to contain the epidemic among MSM, the Curitiba Health Secretariat (SMS), Ministry of Health's Oswaldo Cruz

Foundation (Fiocruz), Federal University of Paraná (UFPR), and Centers for Disease Control and Prevention (CDC) Office in Brazil partnered to launch an implementation science project called "A Hora é Agora—The Time is Now" (AHA). AHA's specific objectives were to increase MSM's access to HTC through a range of testing venues, and to support linkage to care for MSM who tested HIV positive through treatment uptake. AHA's diverse approaches were deliberate in their promotion of integrated services that were free from stigma and discrimination with a focus on recruiting young MSM, aged 16 to 28 years, for services using a dynamic health communications plan.

This article describes the implementation of, and comparison between the number of MSM tested in each strategy, its positivity and linkage; and the social, demographic and behavioral characteristics of MSM accessing different HTC and linkage services. In addition to assessing client characteristics across service delivery platforms, we provide cost data per MSM diagnosed with HIV and per MSM who accepted linkage services to HIV treatment, aimed at improving the understanding of resources required to scale-up testing strategies throughout Brazil.

## Methods

This research was approved by the local (Curitiba) and national ethics committees in Brazil: Ethics Committee of the Municipal Health Secretariat of Curitiba; Ethics Committee of the National School of Public Health Sergio Arouca and the National Ethics Committee. Also, this project was reviewed in accordance with CDC human research protection procedures and was determined to be non-research, public health program activity.

All patients have provided informed written consent to have their data used in the research.

### Target population and subgroups

**Curitiba municipal HTC clinic (COA); Non-Governmental Organization (NGO); HTC Mobile Testing Unit (MTU).** MSM, at least 14 years of age, who signed the informed consent form and MSM, at least 14 years of age, recently diagnosed HIV-positive who accepted linkage support and lived in Curitiba.

**Self-testing.** Men, at least 18 years of age, living in Curitiba.

### Setting and location

Curitiba, Parana, Brazil.

### Study perspective

It´s a cross-sectional analysis of programmatic data collected for each of AHA's HTC strategies as well as from the MSM who accepted linkage support. Those excluded from analysis include transgender women, and MSM who did not sign the informed consent form.

The cost study adopted an ingredients-based approach [14], collecting cost data incurred by each element of the AHA´s HTC strategies.

### Description of the strategies

AHA's strategies were implemented at both fixed and mobile testing outlets. Fixed service sites were the COA and the local lesbian, gay, bisexual and transgender (LGBT) NGO "Grupo Dignidade". In addition, AHA deployed a mobile outlet–MTU operating after business hours. HIV-positive MSM diagnosed at COA, the NGO and the MTU were offered the support of a trained professional ("*linkador*", roughly translating as "linker") to facilitate enrollment into

**Table 1. Structure of AHA voluntary counseling and testing (HTC) services—Curitiba/PR/Brazil—03/01/2015 to 03/31/2017.**

| Strategy | Team [1] | Hours of Operation | HIV / AIDS services offered by the strategy | Differences in the flow of procedures |
|---|---|---|---|---|
| MTU | 2 advisors | Fridays (from 6 p.m. to 10 p.m.) and Saturdays (5 p.m. to 9 p.m.) | Guidance on HIV prevention and rapid testing | *Linkers* receive and fill out the registration forms to apply Informed Consent Form |
| | 2 collectors | | | |
| | 2 peer educators | | | |
| | 2 *linkers* | | | |
| NGO | 1 advisor | Tuesdays, Wednesdays and Thursdays, from 6:00 p.m. to 10:00 p.m. | Guidance on prevention and rapid HIV testing, rights and LGBT citizenship | *Linkers* receive and fill out the registration forms to apply Informed Consent Form |
| | 1 collector | | | |
| | 2 peer educators | | | |
| | 1 *linker* | | | |
| COA | 2 advisors | Monday to Friday (8 am to 6 pm) [2] | Guidance on prevention, HIV testing, syphilis, viral hepatitis, PEP, UDM [3], medical care, vaccination | The COA's front desk does approach, and forwards to counselor. Linker is activated only in case of reagent result. |
| | 2 collectors | | | |
| | 2 peer educators | | | |
| | 3 linkers | | | |

Source: AHA Project.

[1]The size and configuration of the local teams changed throughout the implementation to accommodate user demand. The representation is the most frequent during the two years of implementation.

[2]Although the formal working time of the service is this, users are advised to arrive up to one hour before the closing hours of the activities to perform the test, being claimed by the COA team to be the minimum time for the procedure.

[3]UDM: Medicine Dispensing Unit.

Curitiba's health system for doctor's appointments, exams, and antiretroviral treatment (ART). AHA's most innovative approach was the self-testing program that delivered HIV self-test kits using an Internet-based platform (https://www.ahoraeagora.org/). Curitiba males meeting eligibility criteria (at least 18 years of age, living in Curitiba) were able to request self-administered, oral-fluid tests for home delivery or anonymous pick-up in pre-established outlets. Each of the HTC strategies were advertised online, in physical venues mainly frequented by men, virtual gay meeting sites, and in person through a multi-layered communications strategy.

A team of trained professionals was assigned to each outlet according to the venue's characteristics and needs (Table 1). Self-testing required a logistics team to pack, mail and distribute self-test kits. Another team mentored, supervised, trained, and verified activities developed by the linkers. Finally, a monitoring and evaluation team oversaw data collection, conducted data quality assessments, and led scientific and programmatic reporting. AHA professionals were selected competitively and received training in project goals and objectives, the Curitiba HIV epidemic, stigma and discrimination, and data collection and analysis.

The design of AHA's self-testing strategy is described in a stand-alone study protocol that covered issues specific to the introduction of HIV self-testing (HIVST) in Brazil and the establishment of a virtual platform. MSM accessing the web-based platform were able to request HIV self-testing kits anonymously and to return their test results only if they were comfortable doing so. Users with a reactive self-test result were encouraged to obtain a confirmatory, finger-prick rapid test at COA. The self-testing strategy is not being compared with the other three strategies (COA, NGO and MTU), as showed in Tables 1–3.

AHA offered linkage to HIV treatment to MSM who: i) tested HIV positive in one of the project's outlets offering diagnostic testing and those who had a reactive self-test result and

**Table 2. Distribution of tests carried out by the MSM in accordance with the AHA´s strategies—Curitiba/PR/Brazil—03/01/2015 to 03/31/2017.**

| Strategy | Tests performed on MSM | Reactive tests on MSM | | | MSM eligible for linkage (excluding users with previous HIV diagnosis and on ART) [5] | | | MSM eligible and accepted Linkage | | | Linked in less than 90 days [6] | | |
|---|---|---|---|---|---|---|---|---|---|---|---|---|---|
| | | n | %[1] | CI (95%) | N | %[2] | CI (95%) | n | %[3] | CI (95%) | n | %[4] | CI (95%) |
| COA | 749 | 253 | 33.8% | [30.4; 37.2] | 233 | 97.9% | [96.1; 99.7] | 215 | 92.3% | [88.8; 95.7] | 150 | 69.8% | [63.6; 75.9] |
| NGO | 482 | 27 | 5.6% | [3.5; 7.7] | 15 | 68.2% | [48.7; 87.6] | 10 | 66.7% | [42.8; 90.5] | 8 | 80.0% | [55.2; 100.0] |
| MTU | 1,450 | 85 | 5.9% | [4.7; 7.1] | 52 | 67.5% | [57.1; 78.0] | 33 | 63.5% | [50.4; 76.5] | 14 | 42.4% | [25.6; 59.3] |
| Total | 2,681 | 365 | 13.6% | [12.3; 14.9] | 300 | 89.0% | [85.7; 92.4] | 258 | 86.0% | [82.1; 89.9] | 172 | 66.7% | [60.9; 72.4] |

Source: AHA Project.

[1] Denominator is the number of tests performed on MSM.

[2] Denominator is the number of reactive tests, excluding previous HIV diagnosis and or on ART.

[3] Denominator is the population eligible for linkage.

[4] Denominator is the population eligible and accepted linkage.

[5] Statistical tests were used to select proportions (z-test) in pairs for the strategies—COA and NGO; COA and MTU; NGO and MTU—only for NGO and MTU, the result was not significant. According to the test, the proportion of MSM eligible for linkage in the NGO (68.2%) is not statistically significantly higher than the percentage of MSM eligible for linking in the MTU (67.5%).

[6] Statistical tests were performed for the difference of proportions (z test and Fisher's test) in pairs for the strategies—COA and NGO; COA and MTU; NGOs and MTU—only for COA and NGO, the result was not significant. According to the test, the proportion of linked in less than 90 days in the COA (69.8%) is not statistically significantly lower than the proportion of linked in less than 90 days in the NGO (80.0%).

went to COA to confirm, ii) resided in Curitiba; and iii) signed both an informed consent form as well as an authorization for the project team to conduct linkage services (Fig 1). Follow up was performed by a linker and recorded in a excel spreadsheet. The Curitiba medicine logistics database was also used for monitoring purposes. The user was considered "linked to

**Table 3. Sociodemographic and behavior indicators of MSM according to the AHA´s strategies—Curitiba/PR/Brazil—03/01/2015 to 03/31/2017.**

| Characteristics | COA | | NGO | | MTU | | Total—Tests performed on MSM | |
|---|---|---|---|---|---|---|---|---|
| | N | % | n | % | n | % | n | % |
| Total | 749 | 27.9 | 482 | 18.0 | 1,450 | 54.1 | 2,681 | 100.0 |
| SOCIODEMOGRAPHIC | | | | | | | | |
| Median age (Interquartile range) | 25 (22–29) | | 23 (20–28) | | 26 (21–32) | | 25 (21–30) | |
| Ethnicity: White * | 579 | 77.3 | 271 | 56.2 | 873 | 60.2 | 1,723 | 64.3 |
| Education: Some college or higher * | 354 | 47.5 | 282 | 58.5 | 858 | 59.2 | 1,494 | 55.8 |
| BEHAVIORAL | | | | | | | | |
| Condom: Never and Occasionally | 517 | 69.2 | 309 | 64.1 | 953 | 65.7 | 1,779 | 66.4 |
| Has presented STI in the last 12 months: Yes | 146 | 19.7 | 62 | 12.9 | 134 | 9.3 | 342 | 12.8 |
| Has had the HIV test before: Yes | 581 | 78.9 | 367 | 76.1 | 1,075 | 74.1 | 2,023 | 75.8 |
| Learned about the service through communication of the AHA project | 326 | 47.0 | 87 | 18.2 | 110 | 7.6 | 523 | 20.0 |

Source: AHA Project.

Note: the percentages were calculated in relation to the other categories of the variable.

* Statistical tests were performed for the difference of proportions (z-test) in pairs for the strategies—COA and NGO; COA and MTU; NGO and MTU—only for NGO and MTU, the result was not significant.

According to test, the proportion of whites in the NGO (56.2%) is no less than that of the MTU (60.2%) and the proportion of people with some college or higher at the NGO (58.5%) is no less than that of the MTU (59.2%).

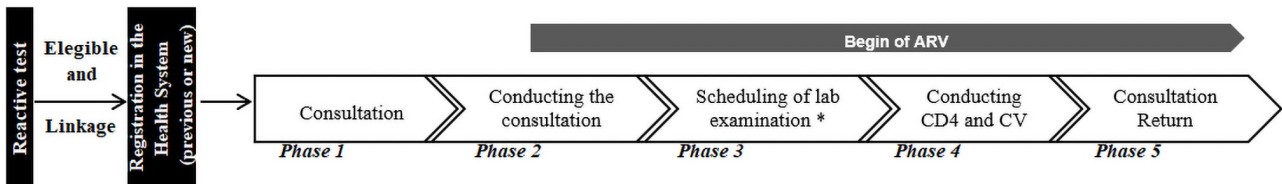

Linkage´s user flow after confirming a reactive HIV test
Source: AHA Project.
* exams include CD4 and Viral Load tests

**Fig 1.**

services" when started on ART. Self-testing users with a positive confirmatory test at COA were also offered linkage services. One-on-one and short messaging services (SMS) were available for up to three months following enrollment in linkage services. At any time along the process, clients were free to decline linkage support while retaining their right to access public health services in Curitiba.

The AHA team developed and implemented a communications plan to increase outreach to gay men and other MSM and encouraged service uptake through virtual tools, e.g., Facebook, Instagram, Twitter, WhatsApp, Grinder, and the AHA website. The project mapped out public venues where MSM gather to socialize [15], including bars, cafés, nightclubs, movie theaters, and public restrooms. This mapping facilitated partnerships with LGBT-friendly businesses, resulting in tailored messaging for each HTC environment and type of client.

The data for the cost study were collected from each of the testing strategies across sites participating in the intervention and evaluation, and across two dimensions for each testing strategy: cost component (personnel, test kits, medical supplies, other supplies, building use, travel & transport, utilities, training, other equipment, vehicles and trailer); and intervention (e.g., testing services, individual counseling, linkage to care, recruitment & communications, and capacity building. Data on allocation of shared resources were collected from key personnel in each testing outlet to aid in the allotment of costs across intervention components and programmatic activities. For cost data analysis, information on the number of people tested, participants that tested positive, and cost per MSM in linkage services. Costs were collected in local currency and the conversion was made over 2016 USD (1USD = R$3.2).

## Time horizon

Data were collected between March 2015 and March 2017 at COA, MTU, NGO and self-testing. For the cost study data were collected retrospective from each testing outlet covering a 12-month period (March 2015 to February 2016).

## Choice of health outcomes

AHA collected socio-demographic and behavioral data through a client registration form required at every AHA testing outlet for inclusion in routine HIV case surveillance and were entered into EpiInfo 7.0 and processed with SPSS 17.0. These data were used periodically by Curitiba's Health Secretariat to flag opportunities to adjust HTC and ART service delivery. Self-testing users filled in the registration form electronically. The system verified inclusion criteria through an anonymous questionnaire, containing socio-demographic and behavioral questions, completed by users prior to placing the self-test kit request.

Data on MSM who tested positive for HIV and accepted linkage support were collected through a standard form and spreadsheet, and compiled for inclusion in health information

systems, including: The National Disease Notification System (SINAN); Lab Test Control System (SISCEL); Drug Logistics Control System (SICLOM); and SMS electronic medical records. Linkage data were entered and processed in Excel 15.0.

Participants' socio-demographic and behavioral characteristics in this analysis were: age group (18 to 29 years, 30 years and more); race/ethnicity (White, non-White); education (completion of high school, attended higher education); condom use (never, occasionally, always); history of sexually transmitted infections (STIs) (yes, no); knowledge of AHA services (friends/partner recommendation, AHA communication materials, health facilities, other); and previous HIV test (yes, no).

Cost data sources were accounting and financial records, equipment inventory, and stocks of materials and consumables. Additional data were collected through key informant interviews aimed at describing the structure and operational flow of project activities. Cost results presented here cover 12 months between March 2015 and February 2016.

The cost analysis covered all AHA´s strategies, including the communications component. Annual cost per testing strategy was estimated for the total number of tests and the total number of HIV-positive tests, coupled with the number of HIV-positive users accepting linkage for each HTC strategy.

## Statistical analysis

In the analyzes, absolute frequencies, percentage distributions and confidence intervals for the percentages were used, as well as unilateral statistical tests for the difference in proportions—z test and/or Fisher test, depending on the number of expected observations per cell.

The objective was to verify whether the differences, greater or lesser, in the proportions were significant by comparing the strategies in the populations "eligible for linking" and "linked in less than 90 days". The tests were carried out comparing the strategies two by two within each of the two populations, as follows: COA x NGO, COA x MTU and NGO x MTU.

Other analyzes used absolute, relative frequencies and median in each of the strategies according to some sociodemographic and behavioral characteristics. For two of the characteristics, whites and some college or higher, unilateral statistical tests were also performed for the difference in proportions—z-test and Fisher's test comparing within the strategies.

## Estimating resources and costs

The idea of the cost analysis was to understand the costs of all the components needed to identify a positive and link them to treatment. All of the activities are interrelated, but they were defined as separate buckets for the process of data collection. While the dollars spent to build capacity for HCW would improve the delivery of testing services or linkage activities, they are very different activities that require different inputs. In order to identify a positive, support for activities within each of those buckets is needed, the survey and enumerators were given instruction on how to differentiate resources that would fall under each activity bucket. Testing services were activities that were directly related to the testing of the individual, such as staff time to perform the test, etc. Linkage activities were activities that were identified as being directly related to linking an HIV positive individual to treat (included costs such as personnel). Capacity building of HCW were any activities that supported the development or delivery of trainings or materials (such as training personnel, material printing, venue costs, per diems to attend training, etc.). Significant effort in survey material and training was spent to ensure items were not double counted and were properly tagged to the correct bucket.

## Results

Over two years of implementation, AHA enrolled 9,268 males into both fixed, mobile testing strategies and self-testing. Results are presented in two parts: 1) fixed and mobile strategies; 2) self-testing.

### Fixed and mobile strategies

Men who reported sex with other men were stratified by AHA testing strategy and by number and proportion of positive test results, eligibility for linkage, and acceptance of linkage support. Table 2 shows 2,681 HIV tests among MSM, of which 365 (13.6%) tested positive. Of these, 28 had either had a previous HIV diagnosis or were already in ART; 300/337 (89%) were eligible for linkage; 258/300 accepted linkage support (86%); 172/258 (66.7%) were linked to services in less than 90 days. In addition, 61/258 MSM (23.6%) were linked to care in more than 90 days; and 25/258 (9.7%) were not linked to services.

According to the linker´s records, the main barriers to delayed initiation of treatment and non-linking are related both to individual users (difficulty to progress in the process, missed appointments and denial of diagnosis); communication difficulties (user does not respond or takes too long to respond), as well as to the care network (delay or difficulties in getting an appointment, and date and time conflicts with the appointment.

Among MSM tested for HIV, the majority did so at the MTU (54.1%), followed by COA (27.9%) and the NGO (18%) (Table 3). COA recorded the highest HIV positivity (33.8%) among the AHA strategies, followed by the MTU (5.9%) and NGO (5.6%). COA also registered the highest proportion of MSM accepting linkage support (97.9%), followed by the NGO (68.2%) and the MTU (67.5%).

Table 3 reports the socio-demographic and behavioral characteristics of AHA participants by strategy. The median age was 25 years old. Two thirds of MSM were white, and 55.8% attended or completed higher education. In general, social and demographic characteristics were similar across strategies, with the exception of COA. As compared to other strategies, COA registered the higher percentage of white MSM (77.3%) and the lowest percentage of MSM who had some college or higher (47.5%).

The majority (66.4%) of MSM reported irregular condom use and 12.8% reported STI diagnosis in the past 12 months.). Over three quarters (75.8%) reported at least one previous HIV test. The MTU had the lowest proportion of users with STI diagnosis in the past 12 months (9.3%). The highest percentage (78.9%) of users with at least one previous HIV test was observed at COA.

### Internet-based HIV self-testing

A total of 7,352 HIV self-tests were requested over 24 months of which 4,752 (65%) were delivered to the home, or picked up at the pharmacy. The reason for this difference is mainly that people requested to pick up the kits at the pharmacy but it did not occur (57%); also, there were missing information in the addressee field (e.g. ZIP code) for the home kits deliver, and cancelation of requests. The majority (72%) of test kits distributed were mailed to the address indicated by the user, while 27% were picked up at the government-run pharmacy [16].

The availability of confirmatory testing and health system navigation for those who self-reported a positive screening (oral) test in the self-testing platform and mobile application were critical components of the comprehensive project. Although not mandatory, 34 individuals voluntarily reported a reactive test result on the project website (De Boni et al., 2018), and 44 sought confirmatory testing in COA. However, in order to guarantee confidentiality, there is no way to know if these users are the same ones who reported the result on the platform. Of

**Table 4. Results of the cost-analysis according to the number and cost of the tests according to the AHA´s strategies—Curitiba/PR/Brazil—03/01/2015 to 02/28/2016.**

| Indicators | COA | NGO | MTU | Self-Testing |
|---|---|---|---|---|
| Number of tests | 244 | 436 | 2,749 | 2,679 |
| Number of positive tests | 106 | 25 | 86 | N/A |
| % of positive tests | 43% | 6% | 3% | N/A |
| Number of tests in MSM | 238 | 228 | 860 | 1,910 |
| Number of positive tests in MSM | 104 | 17 | 69 | N/A |
| % of positive tests in MSM | 44% | 7% | 8% | N/A |
| Number of MSM with positive tests accepting linkage | 77 | 6 | 16 | N/A |
| % of MSM with positive tests who accepting linkage | 74% | 35% | 23% | N/A |
| **Cost per unit (in dollar)** | **COA** | **NGO** | **MTU** | **Self-Testing** |
| Test | 377 | 190 | 43 | 176 |
| Positive test | 867 | 3,311 | 1,370 | N/A |
| Test in MSM | 386 | 363 | 137 | 247 |
| Positive test in MSM | 884 | 4,870 | 1,708 | N/A |
| MSM with positive tests accepting linkage | 1,194 | 13,798 | 7,365 | N/A |

Source: AHA Project.

*Costs were collected in local currency and the conversion was made over 2016 USD (1USD = R$3.2).

those who went to COA, 42 where eligible and 40 accepted linkage support to HIV services. Thirty-two (80%) of those were linked to service in less than 90 days (average 51 days).

## Cost analysis

The costs per test among participants by AHA strategy varied from US$ 43 for the MTU to US$ 377 for the COA (Table 4). Costs per tested positive ranged from US$ 867 at COA to US$ 3,311 at the NGO. Since it was not possible to have accurate information on positive tests for self-testing, this strategy was excluded from cost analysis of HIV-positive results. Costs per MSM tested ranged from US$ 137 to US$ 386, with mobile testing reporting the lowest cost and COA reporting the highest. The cost per positive MSM ranged from US$ 884 for COA to US$ 4,870 for the NGO, and the cost per MSM that accepted linkage to HIV services ranged from US$ 1,194 for COA to US$ 13,798 for NGO.

The distribution of cost components per HIV test was concentrated in personnel costs ranging from 42% in self-testing to 70.5% in the COA to (Table 5). Per the table below, the strategies are similar in cost structure per individual tested, with the exception of self-testing, which had a much greater share of costs allocated to HIV test kits (21.1%), other supplies (14.7%), and travel and transportation (10.2%). Because self-testing relies on mailed tests, it is less labor intensive for healthcare providers, and costs depend mostly on the price of test kits, and shipping fees per individual screened. It is worth mentioning that the costs associated with in-person confirmatory screening is included in the COA unit.

Linkage-related activities at COA accounted for 37.9% of the unit cost per test, followed by counseling, at 28.7% (Table 6). The NGO had linkage activities accounting for 27.6%, recruitment and communications accounting for 22.7%, and 22.4% falling within counseling activities. Mobile testing also reported a relatively even distribution across intervention components, with linkage at 26.9%, testing services at 23.5%, and counseling at 20.7%. Self-testing presented a different cost distribution when compared to the other strategies, with testing services accounting for 49.5% of the cost, owing to the high cost of importing test kits.

**Table 5. Distribution of cost components according to the AHA´s strategies—Curitiba/PR/Brazil.**

| Cost componenents | COA | ONG | MTU | Self-Testing |
|---|---|---|---|---|
| Personnel—Salaries | 62.5% | 70.5% | 66.8% | 41.7% |
| Personnel—Top-Off | 13.8% | 5.2% | 13.1% | 5.9% |
| HIV Test Kits | 0.2% | 0.3% | 1.8% | 21.1% |
| Medical Supplies | 0.2% | 0.3% | 2.3% | 0.1% |
| Other Supplies | 0.1% | 1.9% | 1.0% | 14.7% |
| Building Use | 7.6% | 5.1% | 1.4% | 0.8% |
| Travel & Transport | 6.2% | 6.9% | 2.9% | 10.2% |
| Utilities | 3.6% | 3.5% | 4.4% | 3.5% |
| Training | 5.4% | 6.0% | 4.2% | 1.7% |
| Other Equipement | 0.3% | 0.3% | 0.1% | 0.4% |
| Vehicles and Trailers | 0.0% | 0.0% | 1.9% | 0.0% |
| **Total** | **100.0%** | **100.0%** | **100.0%** | **100.0%** |

Source: AHA Project.

Recruitment and communication activities accounted for 46.5% of the cost. This was particularly high due to initial investments needed in the outreach campaign and website development. Self-testing´s longterm operational costs are expected to lower with time, as this initial investment will not be needed on an annual basis.

## Discussion

AHA successfully provided a variety of HIV testing options to a population of MSM, nearly a quarter of whom had never previously tested. The combination of AHA strategies yielded an HIV prevalence of 13.6% in a concentrated HIV epidemic marked by growing infections among young MSM with little knowledge of HTC, who identify stigma and discrimination as barriers to accessing HIV/AIDS care [17, 18]. According to BRITO et al. (2015) the response to increasing HIV testing rates among MSM requires service availability in friendly, stigma- and discrimination-free facilities. TERTO (2015) states that comprehensive programs involving easy access to rapid testing, health care and ART, in addition to adherence and retention-enhancing mechanisms, are essential for the success of combination prevention strategies.

Between March 2015 and March 2017, AHA performed 2,681 HIV tests among MSM across in-person strategies. The absolute number of tests and the number of MSM tested was greater in the MTU. AHA's strategies engaged mostly white MSM, with average age of 25 years, who had some college or higher education, with at least one previous HIV test and no

**Table 6. Distribution of intervention component according to the AHA´s strategies—Curitiba/PR/Brazil.**

| Intervention Component | COA | ONG | MTU | Self-Testing |
|---|---|---|---|---|
| Testing Services | 12.7% | 15.9% | 23.5% | 49.5% |
| Individual Counseling | 28.7% | 22.4% | 20.7% | 1.1% |
| Linkage to Care | 37.9% | 27.6% | 26.9% | 2.6% |
| Recruitment & communication | 15.8% | 22.7% | 18.0% | 46.5% |
| Capacity Building of HCWs | 4.9% | 11.4% | 10.9% | 0.4% |
| **Total** | **100.0%** | **100.0%** | **100.0%** | **100.0%** |

Source: AHA Project.

self-reported recent STI diagnosis. This cut is in line with the profile of Curitiba's residents [19]. The higher prevalence among testers at COA may have been inflated by MSM who screened reactive in the self-testing strategy and looked for a confirmatory test at COA.

According to a 2015 study in Curitiba [20], HIV risk related to sexual exposure is more frequent among young people, who must be mobilized to engage in HIV testing early on in their sexual lives. Outreach to, and awareness-raising among this group presents greater challenges, requiring diversified and expanded service access strategies. AHA sought to meet these requirements through innovative, stigma- and discrimination-free services in numerous HTC outlets available in alternative shifts, and active and supportive linkage of positive cases to care and treatment.

The introduction of linkage services in testing sites builds a much-needed bridge between diagnosis and treatment for stigmatized MSM. Linkage acceptance rates were high (86%) among eligible MSM. The linkers provided key assistance and support throughout the linkage-to-care process using approaches, language and communication tools that appeal to gay men and MSM, besides ensuring an environment that is free from stigma and discrimination for MSM in the public health care network. Linkers in-depth understanding of the health system mechanics and operations assisted in successfully link users to services. Capacity building was fundamental to build that knowledge. Another positive outcome was high acceptance of AHA-introduced technological innovations in HIV/AIDS testing and care. In particular, Self-Testing emerged as a feasible strategy to increase MSM access to self-testing through virtual means, as the demand far exceeded AHA's foreseen distribution of oral self-test kits.

Self-testing ranked second in terms of cost by MSM tested and has proven highly effective in reaching gay men and MSM, being the strategy that had the largest number of tests distributed for this key population (n = 1,910). It is important to highlight that the cost analysis was done in the first year of the project, when a low quantity of tests was purchased at higher cost. HIV self-tests costs have dropped since project inception with economies of scale. Internet-based HIV self-testing has the potential to become a public health strategy for HIV/AIDS control in concentrated epidemics. In addition, it may be easily adapted to other settings and key populations, e.g., sex workers. With regards to positivity, COA ranked first and consequently had the lowest cost for positive test in MSM, what could be explained by the self-testing confirmatory tests that was referred to that unit.

AHA activities also contributed to expanded knowledge about existing health services and facilities in Curitiba by virtually disseminating information about addresses, working hours and services available, and building the capacity of selected facilities to provide quality services to MSM. Lastly, AHA strengthened Curitiba's HIV/AIDS surveillance system and supported HIV/AIDS service decentralization to the basic health care network.

## Conclusions

Although challenges persist in sustaining alternative HTC and HIVST strategies, AHA implemented and assessed the comparative feasibility and acceptability of various novel approaches to encourage HIV testing uptake among MSM in Curitiba. Innovative strategies implemented by AHA, and particularly the self-testing and linkage-to-care strategies were well accepted by users and providers.

Sustaining these services requires restructuring strategies to reduce associated costs and promote public health network ownership of procedures, processes and functions. This task demands flexibility in the absence of mechanisms and/or resources to include new functions in traditional health services, including peer educators and linkers.

Despite implementation barriers and challenges inherent to an intervention the size and complexity of AHA, the project achieved considerable success. Lessons learned will assist others in adapting these strategies to other settings. AHA's contribution to Brazil's ability to respond specifically to an epidemic concentrated among MSM is considered as AHA's main achievement.

## Acknowledgments

Aristides Barbosa Júnior, Juliane Villela, David Harrad, Adriane Wollmann, Elina Sakurada, Roberto de Jesus, Leonardo Linconl; Raquel De Boni, Renato Lima, Renato Girade Corrêa, Raquel Lima Miranda; the peer educators, the *linkators* and the whole team.

## Author Contributions

**Conceptualization:** Marly Marques da Cruz, Vanda Lúcia Cota, Nena Lentini, Trista Bingham, Gregory Parent, Liza Regina Bueno Rosso.

**Data curation:** Marly Marques da Cruz, Vanda Lúcia Cota, Gregory Parent, Raquel Maria Cardoso Torres, Cristiane Yumi Nakamura.

**Formal analysis:** Marly Marques da Cruz, Vanda Lúcia Cota, Gregory Parent, Solange Kanso, Raquel Maria Cardoso Torres, Cristiane Yumi Nakamura.

**Funding acquisition:** Marly Marques da Cruz, Vanda Lúcia Cota, Nena Lentini, Ana Carolina Faria e Silva Santelli.

**Investigation:** Marly Marques da Cruz, Vanda Lúcia Cota.

**Methodology:** Marly Marques da Cruz, Vanda Lúcia Cota, Gregory Parent, Solange Kanso.

**Project administration:** Marly Marques da Cruz, Vanda Lúcia Cota, Liza Regina Bueno Rosso, Bernardo Almeida.

**Resources:** Marly Marques da Cruz, Vanda Lúcia Cota.

**Supervision:** Marly Marques da Cruz, Vanda Lúcia Cota, Nena Lentini, Trista Bingham, Bernardo Almeida.

**Validation:** Marly Marques da Cruz, Vanda Lúcia Cota, Gregory Parent.

**Visualization:** Marly Marques da Cruz, Vanda Lúcia Cota, Nena Lentini, Trista Bingham.

**Writing – original draft:** Marly Marques da Cruz, Vanda Lúcia Cota, Nena Lentini, Solange Kanso.

**Writing – review & editing:** Marly Marques da Cruz, Vanda Lúcia Cota, Nena Lentini, Trista Bingham, Gregory Parent, Solange Kanso, Liza Regina Bueno Rosso, Bernardo Almeida, Raquel Maria Cardoso Torres, Cristiane Yumi Nakamura, Ana Carolina Faria e Silva Santelli.

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
