## [Decision Letter · Decision Letter 0]

6 Jan 2021

PONE-D-20-13677

Comprehensive approach to HIV/AIDS testing and linkage to treatment among men who
have sex with men in Curitiba, Brazil

PLOS ONE

Dear Dr. Cota,

Thank you for submitting your manuscript to PLOS ONE. After careful consideration, we
feel that it has merit but does not fully meet PLOS ONE’s publication criteria as it
currently stands. Therefore, we invite you to submit a revised version of the
manuscript that addresses the points raised during the review process.

The reviewers felt that a number of issues should be resolved before the manuscript
can be further considered for publication. In particular, several concerns arose
about the presentation of the methodology, the discussion and the reporting
methods/criteria that should be used in the study. The reviewers' comments can be
viewed in full below.

Please submit your revised manuscript by Feb 19 2021 11:59PM. If you will need more
time than this to complete your revisions, please reply to this message or contact
the journal office at plosone@plos.org. When
you're ready to submit your revision, log on to https://www.editorialmanager.com/pone/ and select the 'Submissions
Needing Revision' folder to locate your manuscript file.

If you would like to make changes to your financial disclosure, please include your
updated statement in your cover letter. Guidelines for resubmitting your figure
files are available below the reviewer comments at the end of this letter.

We look forward to receiving your revised manuscript.

Kind regards,

Natasha McDonald, PhD

Associate Editor

PLOS ONE

Journal Requirements:

2. ‘In ethics statement in the manuscript and in the online submission form, please
provide additional information about the patient records used in your retrospective
study. Specifically, please ensure that you have discussed whether all data were
fully anonymized before you accessed them and/or whether the IRB or ethics committee
waived the requirement for informed consent. If patients provided informed written
consent to have data from their medical records used in research, please include
this information.

"All authors declare that they have no significant competing financial, professional,
or personal interests that might have influenced the performance or presentation of
the work described in this manuscript."

Reviewers' comments:

Reviewer's Responses to Questions

**Comments to the Author**

1. Is the manuscript technically sound, and do the data support the conclusions?

Reviewer #1: Partly

Reviewer #2: Partly

2. Has the statistical analysis been performed
appropriately and rigorously? 

Reviewer #1: Yes

Reviewer #2: No

3. Have the authors made all data underlying the
findings in their manuscript fully available?

Reviewer #1: Yes

Reviewer #2: Yes

4. Is the manuscript presented in an intelligible
fashion and written in standard English?

Reviewer #1: Yes

Reviewer #2: No

5. Review Comments to the Author

Reviewer #1: This is a useful study looking at various HIV testing/screening methods
in Brazil. However, additional clarification around each strategy and the
methodology would help the reader more accurately assess the effectiveness of each.
The economic evaluation could be enhanced.

Introduction

Paragraph starting with line 101:

The incidence rates cited are now 6 years old. How have trends changed more recently
with the introduction of PrEP?

Methods

The methods would be more clear if the three comparisons were stated straight away.
It is not clear what is being compared. From table 1, it appears that the MTU is
being compared with fixed services at the COA and NGO. However, the text then
introduces self-testing. How does the self-testing fit? Is it a separated comparison
group or is it a function of one or all of the aforementioned groups?

Other relevant information that needs to be more clear: did all strategies occur at
the same time, or were they simultaneous? Did all allow all ages? Did all recruit
MSM equally, or did particular strategies target a more general LGBT audience?

Cost study-did the cost study cover different dates/time period than the testing
data? If not, please clarify (the dates appear to overlap in table 4, but this is
not mentioned in the methods). If so, I suggest revising to have costs and test
dates align.

Results

Self-testing

Was there an attempt to record the results from these test?

Discussion

The discussion about cost seems to be missing text? Line 425 introduces this topic by
saying that self-testing is ranked second. There is no other discussion of costs.
This should be added. Also, the conclusion that self-testing is effective is not
supported by the data in this analysis since we don’t know how many people actually
used the tests that were delivered.

Reviewer #2: The paper must be substantially improved in order to be published.
Please see specific the comments in the attach file.

The paper must follow some EQUATOR guidelines; in my opinion the CHEERS is the most
adequate.

6. PLOS authors have the option to publish the peer
review history of their article (what does this mean?). If published, this will
include your full peer review and any attached files.

If you choose “no”, your identity will remain anonymous but your review may still be
made public.

**Do you want your identity to be public for this peer review?** For
information about this choice, including consent withdrawal, please see our
Privacy Policy.

Reviewer #1: No

Reviewer #2: No

---

## [Author Response · Author response to Decision Letter 0]

19 Feb 2021

Reviewer #1: 

This is a useful study looking at various HIV testing/screening methods in Brazil.
However, additional clarification around each strategy and the methodology would
help the reader more accurately assess the effectiveness of each. The economic
evaluation could be enhanced.

Introduction

Paragraph starting with line 101: The incidence rates cited are now 6 years old. How
have trends changed more recently with the introduction of PrEP?

Response: the reference has been updated.

Methods

The methods would be more clear if the three comparisons were stated straight away.
It is not clear what is being compared. From table 1, it appears that the MTU is
being compared with fixed services at the COA and NGO. However, the text then
introduces self-testing. How does the self-testing fit? Is it a separated comparison
group or is it a function of one or all of the aforementioned groups?

Response: this section was re-written. The self -testing strategy is not being
compared with the other three strategies (COA, NGO and MTU).

Other relevant information that needs to be more clear: did all strategies occur at
the same time, or were they simultaneous? Did all allow all ages? Did all recruit
MSM equally, or did particular strategies target a more general LGBT audience?

Response: please see time horizon and target population. All strategies targeted the
same population equally – MSM, however the e-testing (self-testing) targeted MSM
aged 18 and above and the other three strategies, target MSM aged 14 and above.

Cost study-did the cost study cover different dates/time period than the testing
data? If not, please clarify (the dates appear to overlap in table 4, but this is
not mentioned in the methods). If so, I suggest revising to have costs and test
dates align.

Response: please see time horizon. The cost study collected retrospective from each
testing outlet covering a 12-month period (March 2015 to February 2016).

Results

Self-testing

Was there an attempt to record the results from these tests?

Response: the results were collected and reported however there is no comparison with
the other three strategies

Discussion

The discussion about cost seems to be missing text? Line 425 introduces this topic by
saying that self-testing is ranked second. There is no other discussion of costs.
This should be added. Also, the conclusion that self-testing is effective is not
supported by the data in this analysis since we don’t know how many people actually
used the tests that were delivered.

Response: this paragraph has been re-written

Reviewer #2: 

The paper must be substantially improved in order to be published. Please see
specific the comments in the attach file.

The paper must follow some EQUATOR guidelines; in my opinion the CHEERS is the most
adequate.

General Observations

1.I had some difficulties in order to understand the main objective of the paper and
its specific objectives; in the abstract there are objectives defined but related to
the project AHA, not to this paper; then, in the introduction (last paragraph) it is
stated another different objective. Also these objectives are not related to the
Title. 

Reading the paper, I get the impression that the main objective of this paper is to
compare the cost-effectiveness of several interventions/strategies used in Curitiba
to expand access to HIV rapid testing in MSM and to improve the linkage of HIV
positive MSM to health services and treatment. If it is the case, then the title
should be different, namely it must reflect the type of approach used by the authors
(health economic evaluation/ cost/ cost-effectiveness, etc..).

Response: revisions made

2. I had some difficulties in finding the most adequate Equator Guideline to review
the article; however, I think that it should be the CHEERS Checklist- Items to
include when reporting economic evaluations of health interventions. Consequently,
the paper must be revised according to this Guideline.

Response: revisions made

Specific Commentaries

Abstract

Introduction. The authors must write in a very clear way the aim and the specific
objectives of the paper (and not the objectives of the AHA)

Methods. Specify the type of study, the participants and statistical tests used to
compare participant’s characteristics between groups. 

Results. They should respond to specific objectives; it is not the case.

Response: the abstract was rewritten.

Introduction

In line 117 it will be interesting to have some figures about the prevalence/
incidence of HIV in Curitiba population if possible by sex and age and compared with
the whole country.

Response: information added to the manuscript

In line 130, aim and specific objectives must be clearly defined (in order to be then
compared with the results). If possible, specify the research question.

Response: the aims and objectives were rewritten. 

Methods

This section must be reorganized, ideally following the structure bellow:

• Target population and subgroups

• Setting and location 

• Study perspective and its relation to the costs being evaluated.

• Description of the interventions or strategies being compared and

state why they were chosen.

• Time horizon 

• Choice of health outcomes

• Measurement of effectiveness

• Estimating resources and costs

• Statistical Analysis (with the tests used) and why they were chosen

Response: This section has been re-written

In lines, 138-141, separate the type of study, from the variables.

Response: it was re-written

In lines 235-241 the description of variables is not a statistical analysis. 

Response: correction made – description was moved to “Choice of health outcomes”
within methods section

In line 241 the authors stated they used frequencies and medians; its ok; however, in
the results (footnotes in Tables 2 and 3) they refer to statistical tests; these
tests must be described in the statistical analysis section.

Response: has been rewritten and is in the statistical analysis section within
methods section

Lines 242-245 are not related to statistical analysis.

Response: correction made – description was moved to “Choice of health outcomes”
within methods section

Results

The Results section must follow CHEERS guidelines and must be coherent with the
specific objectives. 

Response: revision was made in both sections in order to adjust coherency with the
objectives. We did not follow Cheers guidelines as this paper is not an economic
evaluation, even though there is cost analysis included in it.

Table 2 in footnote 6 it is stated that: Statistical tests were performed for the
difference of proportions (z test and Fisher's test) in pairs for the strategies -
COA and NGO; COA and MTU; NGOs and MTU - only for COA and NGO, the result was not
significant.

The sentence: the result was not significant is not adequate; what result? What is
being compared? What is the significant level? 

Tests must be described in Methods section; and in the results we have to understand
which result was not significantly different from another one. Because we are
comparing groups. More rigour is needed.

Response: the footnote was rewritten and the tests performed were described in the
statistical analysis section within methods

Table 3. Footnote. Same observation.

Response: the footnote was rewritten and the tests performed were described in the
statistical analysis section within methods

Table 4. The title is related to COST-EFFECTIVENESS. It is the first time we are
faced with this expression that has a very powerful meaning in economic evaluations
of health interventions

The authors must explain in the Methods section what type of cost-effectiveness
approach they are using and why?

Response: this section has been re-written with appropriated wording

Figures 2 and 3 are not very clear from the graphical point of view. I think that
tables with figures are more frequently used in this context.

Response: the figures were transformed into tables

---

## [Editor Report · Decision Letter 1]

18 Mar 2021

PONE-D-20-13677R1

Comprehensive approach to HIV/AIDS testing and linkage to treatment among men who
have sex with men in Curitiba, Brazil

PLOS ONE

Dear Dr. Cota

After careful consideration, we feel that major revisions suggested by the reviewers
have been addressed. However, there are still minor changes that need to be
considered before publication.

Methods Section, Statistical Analysis

The first paragraph must be corrected as suggested below:

"In the analyzes, absolute frequencies, ......, as well as unilateral statistical
tests for the difference in proportions - *z test and/or Fisher test,
depending on the number of expected observations per cell*."

In the third paragraph don’t include table numbers (Table 3) because these tables are
in the results.

Results section

Footnotes in Table 3 need to be revised.

Footnote 5- According to test, the proportion of the NGO (68.2%) is not
*significantly* greater than that of the MTU (67.5%).

Footnote 6- According to the test, the proportion of the COA (69.8%) *is not
significantly* lower than that of the NGO (80.0%).

In table 3 replace average age by *median age* to be consistent with
the sentence in the paragraph below the table.

We look forward to receiving your revised manuscript.

Kind regards,

Maria do Rosário Oliveira Martins, Ph.D

Academic Editor

PLOS ONE
---

## [Author Response · Author response to Decision Letter 1]

23 Mar 2021

Methods Section, Statistical Analysis: The first paragraph must be corrected as
suggested below:

 "In the analyzes, absolute frequencies, ......, as well as unilateral statistical
tests for the difference in proportions - z test and/or Fisher test, depending on
the number of expected observations per cell."

Response: Changes made.

In the third paragraph don’t include table numbers (Table 3) because these tables are
in the results.

Response: Changes made.

Results section: Footnotes in Table 3 need to be revised. 

Footnote 5- According to test, the proportion of the NGO (68.2%) is not significantly
greater than that of the MTU (67.5%).

Footnote 6- According to the test, the proportion of the COA (69.8%) is not
significantly lower than that of the NGO (80.0%).

Response: You mentioned table 3, but the notes refer to table 2. Changes made on
table 2.

In table 3 replace average age by median age to be consistent with the sentence in
the paragraph below the table.

Response: Changes made.

---

## [Editor Report · Decision Letter 2]

29 Mar 2021

Comprehensive approach to HIV/AIDS testing and linkage to treatment among men who
have sex with men in Curitiba, Brazil

PONE-D-20-13677R2

Dear Dr. Vanda Cota,

We’re pleased to inform you that your manuscript has been judged scientifically
suitable for publication and will be formally accepted for publication once it meets
all outstanding technical requirements.

Kind regards,

Maria do Rosário Oliveira Martins, Ph.D

Guest Editor

PLOS ONE

Additional Editor Comments (optional):

The authors made all suggested changes.

The paper can be accepted.
---

## [Editor Report · Acceptance letter]

26 Apr 2021

PONE-D-20-13677R2 

Comprehensive approach to HIV/AIDS testing and linkage to treatment among men who
have sex with men in Curitiba, Brazil 

Dear Dr. Cota:

I'm pleased to inform you that your manuscript has been deemed suitable for
publication in PLOS ONE. Congratulations! Your manuscript is now with our production
department. 

Kind regards, 

on behalf of

Professor Maria do Rosário Oliveira Martins 

Guest Editor

PLOS ONE